

# Seasonal Prediction of Winter Haze Days in the North-Central North China Plain

Zhicong Yin [1,2], Huijun Wang [1,2,3]

[1]Collaborative Innovation Center on Forecast and Evaluation of Meteorological Disasters/Key Laboratory of Meteorological
5    Disaster, Nanjing University of Information Science & Technology, Nanjing, China

[2]Nansen-Zhu International Research Centre, Institute of Atmospheric Physics, Chinese Academy of Sciences, Beijing, China

[3]Climate Change Research Center, Chinese Academy of Sciences, Beijing, China

*Correspondence to*: Zhicong Yin (yinzhc@163.com)

**Abstract.** Recently, the winter (December–February) haze pollution over the North-Central North China Plain (NCP) has

10    become severe. By treating the year-to-year increment as the predictand, two new statistical schemes were established using

the multiple linear regression (MLR) and the generalized additive model (GAM) approaches. By analyzing the associated

increment of atmospheric circulation, seven leading predictors were selected to predict the upcoming winter haze days over

the NCP (WHD$_{NCP}$). After cross validation, the root mean square error and explained variance of the MLR (GAM) prediction

model was 3.39 (3.38) and 53% (54%), respectively. For the final predicted WHD$_{NCP}$, both of these models could capture the

15    interannual and interdecadal trends and the extremums successfully. Independent prediction tests for 2014 and 2015 also

confirmed the good predictive skill of the new schemes. The predicted bias of the MLR (GAM) prediction model in 2014

and 2015 was 0.09 (−0.07) and −3.33 (−1.01), respectively. Compared to the MLR model, the GAM model had a higher

predictive skill in reproducing the rapid and continuous increase of WHD$_{NCP}$ after 2010.

## 1.    Introduction

20       In recent years, the North-Central North China Plain (NCP; 34–43ºN, 114–120ºE) has suffered from increasingly severe

winter (December–February) haze pollution (Ding et al. 2014), particularly after persistent heavy fog and haze in January

2013 (Zhang et al. 2014; Zhao et al. 2014). After 2000, the combined effects of a rapid increase in total energy consumption

and the influence of climate change intensified the haze pollution in central North China (Wang et al. 2016). In conditions of





heavy and slowly varying pollutant emissions, the fine particles in the atmosphere reach their saturation levels easily, and the

climate conditions become critical contributors of haze. Some new climatic findings have been helpful seasonal predictors of

winter haze days over the NCP ($WHD_{NCP}$). The East Asian winter monsoon (EAWM) has a significantly negative

relationship with $WHD_{NCP}$ (Yin et al. 2015a; Yin et al. 2015b; Li et al. 2015). By weakening EAWM circulations, negative

Sea Surface Temperature (SST) anomalies over the subtropical western Pacific (SWP) could significantly intensify $WHD_{NCP}$

(Yin et al. 2015c). Furthermore, the decline of preceding autumn (Sep–Nov) Arctic Sea Ice (ASI) has led to favorable

environments for haze, with high static stability and greatly intensified haze pollution in eastern China (Wang et al. 2015).

Although recent studies on the changes in $WHD_{NCP}$ and their associated mechanisms are new and still insufficient, they

support the possibility of seasonal prediction.

   The climate variables in East Asia showed obvious characteristics of tropospheric biennial oscillation (TBO), based on

which, a new interannual increment approach was applied for short-term climate prediction (Wang et al. 2000; Wang et al.

2012). This new approach treated the year-to-year increment of a variable, i.e., the difference between the current and

previous year (DY), as the predictand. Because the DY approach utilized the observed information from the previous year

and the features of TBO, the interannual variation and interdecadal trend could be captured well. In addition, the signals (i.e.,

variance) of the predictors and predictand were both amplified (Huang et al. 2015) and, thus, of benefit to improve the

prediction skill. If the predictive objects (Y), e.g., haze days, were cross-influenced by socio-economic factors and climatic

conditions, the predictand could be represented by Y = YS + YC, where YS and YC are the slowly varying socio-economic

and climatic components, respectively.

$$DY = Y_t - Y_{t-1} = (YS_t + YC_t) - (YS_{t-1} + YC_{t-1}) = (YS_t - YS_{t-1}) + (YC_t - YC_{t-1})$$

where the subscripts *t* and *t-1* indicate the current and previous years, respectively.

   Commonly, the difference in pollutant emissions between current and previous year was very small, resulting

in $(YS_t - YS_{t-1}) \approx 0$, so $DY \approx (YC_t - YC_{t-1})$. To some extent, the $WHD_{NCP}$ DY reflected the fluctuations caused by

climate change. After adding the predicted $WHD_{NCP}$ DY to the observed $WHD_{NCP}$ last year, the interdecadal and

socio-economic components were contained in the final prediction. In prior studies, the DY approach has been used to

explore the prediction of summer rainfall in China (Fan et al. 2008), heavy winter snow activity in Northeast China (Fan et al.



2013), summer Asian-Pacific Oscillation (Huang et al. 2014) and winter North Atlantic Oscillation (Tian et al. 2015).

Furthermore, some variables cross-influenced by socio-economic and climatic factors were predicted successfully using the

DY approach, e.g., rice production in Northeast China (Zhou et al. 2014) and the discoloration day for *Cotinus coggygria*

leaves in Beijing (Yin et al. 2014). Considering the seriously negative impact of winter haze and the substantial need to

predict $WHD_{NCP}$, we made it the goal of this study to apply the DY approach to the seasonal prediction of $WHD_{NCP}$.

The data and methods employed are introduced in section 2. Section 3 describes the predictors and associated

circulations. We apply the DY approach to build the prediction models for $WHD_{NCP}$ in section 4. In this section, the

statistical models are built based on multiple linear regression (MLR) and generalized additive model (GAM). Then,

cross-validation and independent tests are performed to assess the statistical schemes of $WHD_{NCP}$ prediction.

## 2.    Datasets and methods

Monthly atmospheric data, such as geopotential height (Z) and surface temperature (TS), are derived from the National

Centers for Environmental Prediction/National Center for Atmospheric Research (NCEP/NCAR) global reanalysis dataset

with a horizontal resolution of $2.5° \times 2.5°$ from 1979 to 2016 (Kalnay et al. 1996). The monthly mean Extended

Reconstructed SST datasets with a horizontal resolution of $2° \times 2°$ from 1979 to 2016 were obtained from the National

Oceanic and Atmospheric Administration (NOAA) (Smith et al. 2008). ASI extent was calculated from the ASI concentration

data, downloaded from the Hadley Center with a horizontal resolution of $1° \times 1°$ from 1979 to 2016 (Rayner et al. 2003). The

monthly gridded soil moisture data from 1979 to 2016 were downloaded from NOAA's Climate Prediction Center (CPC),

with a horizontal resolution of $0.5° \times 0.5°$ (Huug et al. 2003). The monthly Antarctic Oscillation (AAO) indices from 1979 to

2016 were also obtained from the CPC (Mo et al. 2000).

China ground observations from 39 NCP stations, collected by the National Meteorological Information Center of

China 4 times per day from 1979 to 2016, were used to reconstruct the climatic WHD data (Yin et al. 2015c). Here, haze is

defined as visibility less than a certain threshold and relative humidity less than 90%. After excluding other weather

phenomena affecting visibility, a day with haze at any time is defined as a haze day. Site WHD data were converted into



grids after Cressman interpolation (Cressman, 1959), and then the $WHD_{NCP}$ was computed as the mean value of the gridded data.

In this study, the statistical models were built based on MLR and GAM methods. The MLR approach, a model-driven method, is ultimately expressed as a linear combination of $K$ predictors ($x_i$) that can generate the least error for prediction

of $\hat{y}$ (Wilks 2011). With coefficients $\beta_i$, intercept $\beta_0$ and residual ε, the MLR formula can be described as follows:

$$\hat{y} = \beta_0 + \sum_{i=1}^{K} \beta_i x_i + \varepsilon \qquad (1)$$

The GAM approach is more advanced and was developed from MLR and the generalized linear model (GLM) (Hastie et al. 1990). This data-driven method is particularly effective at handling the complex non-linear and non-monotonous relationships between the predictand and the predictors, whose expressions are replaced by smooth functions (s). Similar to GLM, the dependent variable in GAM can have different probability distributions, such as Gaussian, Poisson, and Binomial,

any of which can be transferred by the link function (g). The GAM can be written in the form:

$$g(\hat{y}) = \beta_0 + \sum_{i=1}^{K} \beta_i s(x_i) + \varepsilon \qquad (2)$$

The normalized datasets from 1979 to 2013 were trained as the basic samples to fit the models, and those from 2014 to 2015 were treated as test data for independent prediction. Thereafter, the root mean standard error (RMSE), mean absolute error (MAE) and explained variance (EV) were calculated for evaluation by simple fitting and leave-one-out cross validation.

**3.    The predictors and associated circulations**

To choose the DY predictors, the correlated DY atmospheric circulations were identified, as shown in Figure 1. The positive phase of the East Atlantic/West Russia (EA/WR; Barnston et al. 1987) and Pacific Japan (PJ; Nitta 1987) patterns and the negative phase of the Eurasia (EU; Wallace et al. 1981) pattern were obvious, and we took the anti-cyclone circulation over North China as an intermediary leading to a more stable atmosphere. The positive anomaly over the NCP



could confine the particles within a thinner boundary layer by suppressing vertical movement and induce an easterly to

weaken the East Asia Jet Stream (EASJ), producing weaker cold air. Meanwhile, the water vapor transportation was also

enhanced, creating favorable conditions for more $WHD_{NCP}$ than in the previous year.

The pivotal local anti-cyclone over the NCP was the most important contributor; we therefore speculated that

pre-autumn TS DY around the NCP should be effective to impact $WHD_{NCP}$ DY. There were significantly negative

correlations between $WHD_{NCP}$ DY and pre-autumn TS DY from the Japan Sea to the Stanovoy Range (35–65$^{o}$N, 130–140$^{o}$E),

the area mean of which was selected as predictor $x_1$ (Figure 2). The correlation coefficient (CC) between $WHD_{NCP}$ DY and

predictor $x_1$ was −0.47, exceeding a 99% confidence level. The circulations associated with predictor $x_1$ (×−1) presented

obvious features of the negative EU and positive PJ patterns (Figure 3), similar to those shown in Figure 1.

The pre-autumn SST anomalies of the Pacific could influence $WHD_{NCP}$ significantly via the air-sea interaction (Yin et al.

2015c). Figure 4 shows the CC between $WHD_{NCP}$ DY and pre-autumn SST DY. The most significant CC was distributed

around the Alaska Gulf (36–56$^{o}$N, 130–170$^{o}$W), and the area-averaged SST DY here was defined as predictor $x_2$, whose CC

with $WHD_{NCP}$ DY was 0.47, above the 99% confidence level. The positive SST DY around the Alaska Gulf closely

correlated with the atmospheric teleconnection patterns, i.e., the positive phases of the EA/WR and PJ and the negative EU

patterns intensified haze pollution over the NCP (Figure 5).

Prior studies have documented that the triple SST pattern was a dominant mode of the northern Atlantic in autumn.

When the pre-autumn SST anomalies were distributed in a "+−+" pattern from south to north, the subsequent EAWM was

stronger, and the surface temperature of North China was lower (Shi 2009). Similarly, the CC between $WHD_{NCP}$ DY and

pre-autumn SST DY of the Atlantic was distributed in a "−+−" pattern (Figure 6). The area-averaged SST DY of the northern

center was defined as predictor $x_3$, whose CC with $WHD_{NCP}$ DY was −0.50, passing the 99% confidence test. The most

obvious DY atmospheric circulations related with predictor $x_3$ (×−1) were the negative EU pattern, whose south center

linked with a subtropical high (Figure 7). The continental high and marine low were both weaker, leading to weaker EAWM

and weaker cold air. The pressure gradient over the east coast of China also resulted in a southerly anomaly, indicating

smaller surface wind and more moisture and resulting in more $WHD_{NCP}$.

ASI decreased dramatically with significant variance and was a significant contributor influencing $WHD_{NCP}$ in eastern





China (Wang et al. 2015; Wang et al. 2016). The CC between pre-autumn ASI DY and $WHD_{NCP}$ DY was calculated (Figure 8) and was significantly positive around Beaufort Sea (73–78°N, 130–165°W). The area-averaged ASI extent DY of Beaufort Sea was selected as the fourth predictor ($x_4$), and its CC with $WHD_{NCP}$ DY was 0.37, above a 95% confidence level. A positive center was located over the Central Siberian and Mongolia Plateau, and negative centers were distributed zonally from southern China to the subtropical Pacific (Figure 9). Thus, the EAJS was weakened by the induced easterly.

Soil moisture is an important factor for seasonal prediction, but only after SST (Guo et al. 2007). The questions with respect to soil moisture were whether pre-summer or autumn soil moisture would be effective for seasonal prediction of $WHD_{NCP}$ DY. The area-averaged pre-autumn soil moisture DY of the Bohai rim (35–42°N, 117–127°E), defined as predictor $x_5$, showed a significantly negative correlation with $WHD_{NCP}$ DY, i.e., the CC was −0.59, exceeding a 99% confidence test (Figure 10). The CC between predictor $x_5$ and Z500 (Z at 500 hPa) was distributed in a similar way as in Figure 1. The

positive EA/WR and PJ phases and the negative EU phase was obvious and led to more $WHD_{NCP}$ than in the previous year (Figure 11). As shown in Figure 12, the pre-summer soil moisture DY in the east of Mongolia (48–52°N, 115–125°E) also had a close relationship with $WHD_{NCP}$ and with $WHD_{NCP}$ DY. The area-averaged soil moisture DY in the east of Mongolia was defined as predictor $x_6$, whose CC with $WHD_{NCP}$ DY was 0.41, above a 95% confidence level. The negative EU pattern could be recognized from the associated atmospheric circulation with predictor $x_6$ (Figure 13), which intensified the haze

pollution over the NCP.

    Recently, some studies documented that Antarctic Oscillation (AAO) could affect the East Asian climate through cross-equatorial flow, e.g., the Somali jet (Fan et al. 2004; Fan et al. 2006; Fan et al. 2007a; Fan et al. 2007b). After the late-1990s, global sea level pressure and Z300 in boreal January were characterized by the concurrence of the Aleutian low and the negative phase of the AAO (Li et al. 2014). We investigated the relationship between $WHD_{NCP}$ DY and Z850 in the

Southern Hemisphere and found that the distribution was remarkably similar to that of the negative phase of AAO (Figure 14). Furthermore, the CC between the Sep–Oct AAO DY and $WHD_{NCP}$ DY was −0.54, exceeding a 99% confidence test. As shown in Figure 15, the positive phases of the EA/WR and PJ patterns were closely correlated with the negative phase of Sep–Oct AAO and were responsible for more $WHD_{NCP}$ than in the previous year. Hence, the Sep–Oct mean AAO index was selected as the last predictor ($x_7$) to forecast the interannual increment of $WHD_{NCP}$.



## 4. The prediction models and validations


In total, seven DY predictors ($x_1, x_2 \dots, and\ x_7$) were chosen to build the seasonal prediction model (SPM) for $WHD_{NCP}$

DY (Table 2). Among the predictors were 21 types of pair combinations, of which only 5 presented significant linear

correlation (Figure 16). The multicollinearity would not be a problem when modeling with the MLR approach. Although the

linear correlation between the predictand and each predictor was significant, the non-linear interaction would also affect the

$WHD_{NCP}$ and should be taken into account. In this section, seasonal prediction models were established using MLR

($SPM_{MLR}$) and GAM ($SPM_{GAM}$) and validated in detail.

The $WHD_{NCP}$ DY showed obvious features of biennial oscillation (Figure 17), illustrating the DY approach was suitable

for its prediction. The $SPM_{MLR}$ of $WHD_{NCP}$ DY was as follows: $DY/10 = -2.774x_1 + 2.582x_2 - 1.631x_3 + 2.528x_4 -$

$2.229x_5 + 2.555x_6 - 1.812x_7$. After cross validation, the $RMSE_{CV}$ of $SPM_{MLR}$ was 3.39 days, and the CC between fitted

and observed $WHD_{NCP}$ DY was 0.73, accounting for 53% of the total variance (Table 2). The percentage of same sign (PSS;

same sign means the mathematical sign of the fitted and observed $WHD_{NCP}$ DY was the same) was 79.4%. The $SPM_{MLR}$

showed good ability to predict the negative and least $WHD_{NCP}$ DY but did not adequately capture the continuous positive

value after 2011 (Figure 17a). The fitted $WHD_{NCP}$ DY from 2011 to 2013 varied similarly to that before 2010 and did not

reflect the rising trend after 2010. As an independent prediction test, the predicted bias in 2014 was 0.09, illustrating good

performance, but the bias in 2015 was larger, i.e., −3.33.

We also applied the GAM approach to build a prediction model that would contain the non-linear relationship with

smooth functions. The $SPM_{GAM}$ of $WHD_{NCP}$ DY was as follows: $DY/10 = -2.164s(x_1) + 2.036s(x_2) - 1.721x_3 +$

$2.588s(x_4) - 2.157s(x_5) + 2.187x_6 - 2.506x_7$. During the simple fitting, the $SPM_{GAM}$ performed very well. The RMSE

was 1.56 days, and the CC between the fitted and observed $WHD_{NCP}$ DY was 0.95. The $SPM_{GAM}$ could fit the minimum (in

2003) and maximum (in 2013), and show the trend well, indicating an advantage to processing the non-linear relationship.

After cross validation, the performance of $SPM_{GAM}$ decreased dramatically, meaning that its stability was worse than that of

$SPM_{MLR}$. The $RMSE_{CV}$ of $SPM_{GAM}$ was 3.38 days and the CC between fitted and observed $WHD_{NCP}$ DY was 0.74,

accounting for 54% of the total variance (Table 2). The PSS of $SPM_{GAM}$ results was 73.5%, which is close to the result from



SPM$_{MLR}$. The SPM$_{GAM}$ also showed good ability to predict the negative and minimum WHD$_{NCP}$ DY and better performance

to fit the maximum in 2013 (Figure 17b). The predicted bias in 2014 and 2015 was −0.07 and −1.01, results that are better

than those from SPM$_{MLR}$. The CC between the bias of SPM$_{MLR}$ and SPM$_{GAM}$ from 1980 to 2013 was 0.83, above a 99.99%

confidence level. If the SPM$_{MLR}$ performed well in some years, the SPM$_{GAM}$ also showed good ability in these years, and

*vice versa*. We speculated that the reason was that some useful factors were not diagnosed and included here.

After adding the predicted WHD$_{NCP}$ DY to the observed information in the previous year, the predicted WHD$_{NCP}$ in the

current year was obtained. For example, the predicted WHD$_{NCP}$ DY in 2012 was added to the measured WHD$_{NCP}$ in 2011,

and the result was the final predicted WHD$_{NCP}$ in 2012. In Figure 18, the simulative WHD$_{NCP}$ anomaly was fitted by

cross-validation from 1980 to 2013 and predicted in 2014 and 2015. For SPM$_{MLR}$ and SPM$_{GAM}$, the CC between the original

(detrended) observed and simulative WHD$_{NCP}$ was 0.89 (0.87) and 0.90 (0.88), respectively. Both of these prediction models

could capture the interannual and interdecadal trend and the extremums. The PSS of the anomalies from the two models was

100%, meaning these two models could predict the sign of WHD$_{NCP}$ anomaly successfully. The SPM$_{GAM}$ could simulate the

abrupt rising trend in 2010 better than SPM$_{MLR}$, which was important for the prediction of recent years.

### 5.    Conclusions and Discussions

In this paper, we treated the WHD$_{NCP}$ DY as the predictand and built two prediction models using the MLR and GAM

approach. In the DY atmospheric circulation, the positive phases of the EA/WR and PJ patterns and the negative phase of the

EU pattern intensified the haze pollution by inducing positive anomalies over the NCP. Finally, seven leading predictors

were selected and are listed in Table 2.

After cross validation, the RMSE$_{CV}$ and explained variance of SPM$_{MLR}$ (SPM$_{GAM}$) was 3.39 (3.38) and 53% (54%). The

PSS of these two prediction models was also similar, i.e., more than 73%. The WHD$_{NCP}$ DY increased rapidly and

persistently after 2010, and the SPM$_{GAM}$ could capture this trend better. For the final predicted WHD$_{NCP}$, both of these

prediction models could capture the interannual and interdecadal trends and the extremums. The PSS of the anomalies from

two models was 100%, and the SPM$_{GAM}$ simulated the abrupt increase in 2010 better than SPM$_{MLR}$. The predicted bias of

SPM$_{MLR}$ (SPM$_{GAM}$) in 2014 and 2015 was 0.09 (−0.07) and −3.33 (−1.01), respectively. Both of these models performed well in independent tests, but the biases of SPM$_{GAM}$ were smaller.

Although these two statistical models performed well during most of the past 3 decades and could predict the WHD$_{NCP}$

in 2014 and 2015 with small biases, they showed disadvantages when simulating the rapid rising trend after 2010. If the SPM$_{MLR}$ performed well in some years, the SPM$_{GAM}$ also showed good ability in these years, and *vice versa*. One possible reason could be that some useful factors were not diagnosed and included here. In this paper, we assumed that the difference in pollutant emissions between current and previous years was very small and that the socio-economic component of WHD$_{NCP}$ varied slowly. Another possible reason might be that in certain years, this pollutant emission proportion varied

rapidly.

**Acknowledgement**

This research was supported by the National Natural Science Foundation of China (Grants: 41421004 and 41210007) and CAS-PKU Partnership Program.

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

**Table and Figure Captions:**

**Table 1**. The RMSE, MAE, CC and EV of MLR and GAM models, and predicted bias for 2014 and 2015. The subscripts "S" and "CV" indicated simple and cross-validation fitting.

**Table 2**. The predictors and their meaning. "CC" indicated the correlation coefficient between predictor and WHD$_{NCP}$ DY from 1980 to 2013.

**Figure 1**. The correlation coefficient (CC) between WHD$_{NCP}$ DY and Z500 DY in winter from 1980 to 2013. The white curves indicate that the CC exceeded the 95% confidence level. A and C represent anti-cyclone and cyclone, respectively.

**Figure 2**. The CC between WHD$_{NCP}$ DY and TS DY in autumn from 1980 to 2013. The shades indicate that the CC

exceeded the 95% confidence level, and the rectangle represents the selected region (35–65$^o$N, 130–140$^o$E) of predictor $x_1$.

**Figure 3**. The CC between predictor $x_1$ (×−1) and Z500 DY in winter from 1980 to 2013. The white curves indicate that the CC exceeded the 95% confidence level. A and C represent anti-cyclone and cyclone, respectively.

**Figure 4**. The CC between WHD$_{NCP}$ DY and Pacific SST DY in autumn from 1980 to 2013. The shades indicate that the CC exceeded the 95% confidence level, and the rectangle represents the selected region (36–56$^o$N, 130–170$^o$W) of predictor $x_2$.





**Figure 5**. The CC between predictor $x_2$ and Z500 DY in winter from 1980 to 2013. The white curves indicate that the CC

exceeded the 95% confidence level. A and C represent anti-cyclone and cyclone, respectively.

**Figure 6**. The CC between WHD$_{NCP}$ DY and Atlantic SST DY in autumn from 1980 to 2013. The shades indicate that the

CC exceeded the 95% confidence level, and the rectangle represents the selected region (50–70$^o$N,30–65$^o$W) of predictor

$x_3$.

**Figure 7**. The CC between predictor $x_3$ (×−1) and Z500 DY in winter from 1980 to 2013. The white curves indicate that

the CC exceeded the 95% confidence level. A and C represent anti-cyclone and cyclone, respectively.

**Figure 8**. The CC between WHD$_{NCP}$ DY and ASI DY in autumn from 1980 to 2013. The shades indicate that the CC

exceeded the 95% confidence level, and the rectangle represents the selected region (73–78$^o$N,130–165$^o$W) of predictor $x_4$.

**Figure 9**. The CC between predictor $x_4$ and Z500 DY in winter from 1980 to 2013. The white curves indicate that the CC

exceeded the 95% confidence level.

**Figure 10**. The CC between WHD$_{NCP}$ DY and SoilM DY in autumn from 1980 to 2013. The shades indicate that the CC

exceeded the 95% confidence level, and the rectangle represents the selected region (35–42$^o$N,117–127$^o$E) of predictor $x_5$.

**Figure 11**. The CC between predictor $x_5$ (×−1) and Z500 DY in winter from 1980 to 2013. The white curves indicate that

the CC exceeded the 95% confidence level. A and C represent anti-cyclone and cyclone, respectively.

**Figure 12**. The CC between WHD$_{NCP}$ DY and SoilM DY in summer from 1980 to 2013. The shades indicate that the CC

exceeded the 95% confidence level, and the rectangle represents the selected region (48–52$^o$N,115–125$^o$E) of predictor $x_6$.

**Figure 13.** The CC between predictor $x_6$ and Z500 DY in winter from 1980 to 2013. The white curves indicate that the CC

exceeded the 95% confidence level. A and C represent anti-cyclone and cyclone, respectively.

**Figure 14.** The CC between WHD$_{NCP}$ DY and Sep–Oct Z850 DY from 1980 to 2013. The white curves indicate that the CC

exceeded the 95% confidence level.

**Figure 15.** The CC between predictor $x_7$ (×−1) and Z500 DY in winter from 1980 to 2013. The white curves indicate that

the CC exceeded the 95% confidence level.





**Figure 16.** Correlogram of the dependent (Y) and independent $(x_1, x_2 ..., and\ x_7)$ variables, whose names were written on the diagonal. The lower panel shows the pie charts of correlation coefficients, representing the values by area and saturation, and showing positive/negative sign by blue/red, respectively. The upper panel shows the scatter plots.

**Figure 17.** The temporal variation of measured (black) $WHD_{NCP}$ DY, MLR (red, a) and GAM (red, b) cross-validation fitted $WHD_{NCP}$ DY from 1980 to 2013. The results for 2014 and 2015 represent the measured (black square) and predicted (red hollow circle) $WHD_{NCP}$ DY.

**Figure 18.** The temporal variation of measured (black) $WHD_{NCP}$ anomaly from 1980 to 2015, MLR (blue) and GAM (red) simulative $WHD_{NCP}$ anomaly, which was composed of cross fitted series from 1980 to 2013 and predicted values in 2014 and 2015.

**Table 1: The RMSE, MAE, CC and EV of MLR and GAM models, and predicted bias for 2014 and 2015. The subscripts "S" and "CV" indicated simple and cross-validation fitting.**

|  | $MLR_s$ | $MLR_{CV}$ | $GAM_s$ | $GAM_{CV}$ |
|---|---|---|---|---|
| **RMSE** | 2.39 | 3.39 | 1.56 | 3.38 |
| **MAE** | 1.75 | 2.37 | 1.10 | 2.58 |
| **CC** | 0.87 | 0.72 | 0.95 | 0.74 |
| **EV** | 76% | 53% | 90% | 54% |
| **Bias$_{14}$** | 0.09 | | −0.07 | |
| **Bias$_{15}$** | −3.33 | | −1.01 | |





**Table 2. The predictors and their meaning. "CC" indicated the correlation coefficient between predictor and WHD$_{NCP}$ DY from**

**1980 to 2013.**

| Predictors | Meaning | CC |
|:---:|:---|:---:|
| $x_1$ | pre-autumn TS DY from Japan Sea to Stanovoy Range | −0.47 |
| $x_2$ | pre-autumn SST DY around Alaska Gulf | 0.47 |
| $x_3$ | pre-autumn SST DY to the south of Greenland | −0.50 |
| $x_4$ | pre-autumn ASI extent DY of Beaufort Sea | 0.37 |
| $x_5$ | pre-autumn soil moisture DY of the Bohai rim | −0.59 |
| $x_6$ | pre-summer soil moisture DY in the east of Mongolia | 0.41 |
| $x_7$ | Sep–Oct AAO index DY | −0.54 |


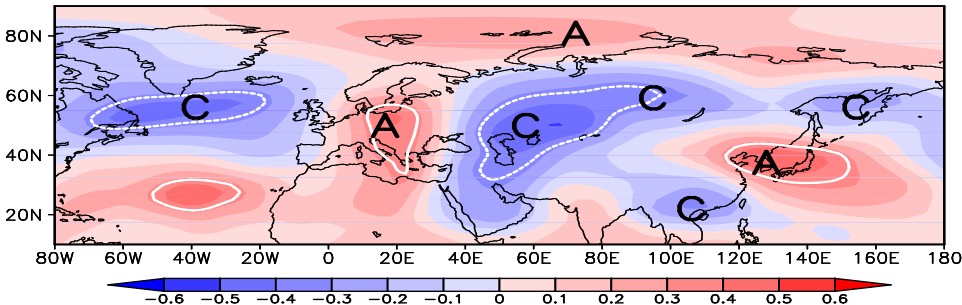

**Figure 1. The correlation coefficient (CC) between WHD$_{NCP}$ DY and Z500 DY in winter from 1980 to 2013. The white curves**

**indicate that the CC exceeded the 95% confidence level. A and C represent anti-cyclone and cyclone, respectively.**




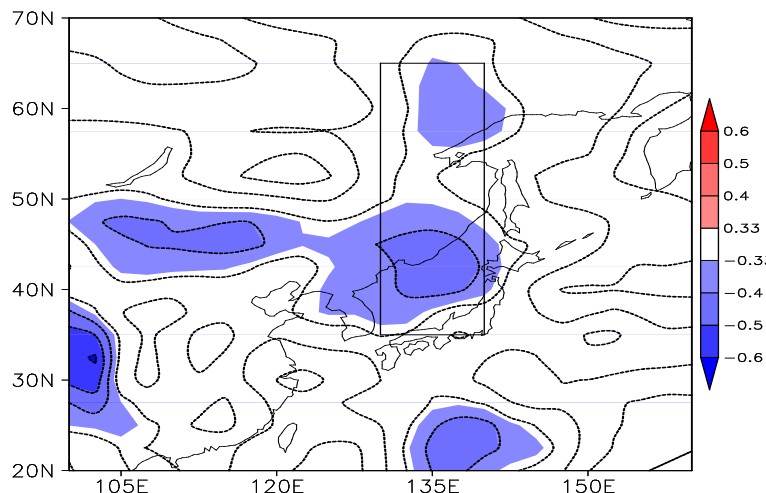

**Figure 2. The CC between WHD$_{NCP}$ DY and TS DY in autumn from 1980 to 2013. The shades indicate that the CC exceeded the 95% confidence level, and the rectangle represents the selected region (35–65$^o$N,130–140$^o$E) of predictor $x_1$.**

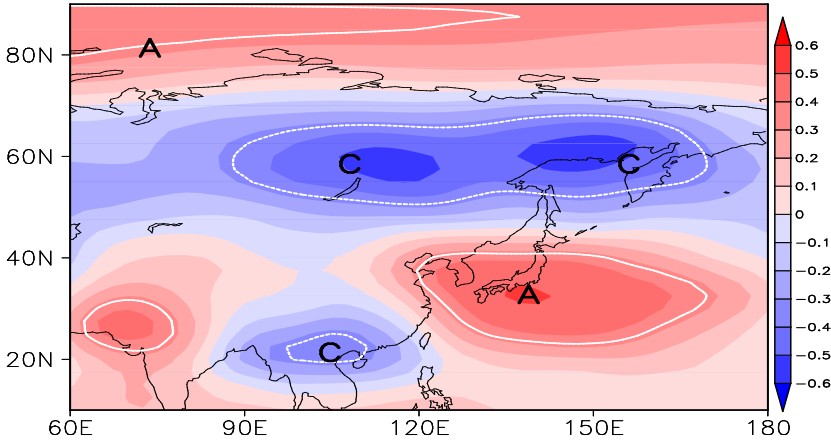

**Figure 3. The CC between predictor $x_1$ (×−1) and Z500 DY in winter from 1980 to 2013. The white curves indicate that the CC exceeded the 95% confidence level. A and C represent anti-cyclone and cyclone, respectively.**



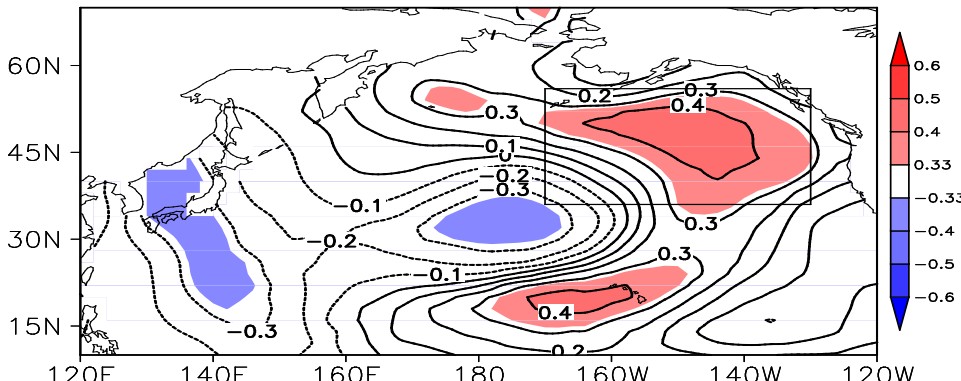

**Figure 4. The CC between WHD$_{NCP}$ DY and Pacific SST DY in autumn from 1980 to 2013. The shades indicate that the CC exceeded the 95% confidence level, and the rectangle represents the selected region (36–56$^o$N,130–170$^o$W) of predictor $x_2$.**

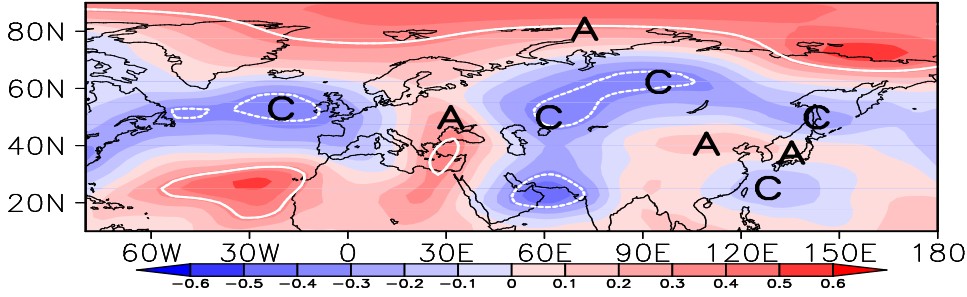

**Figure 5. The CC between predictor $x_2$ and Z500 DY in winter from 1980 to 2013. The white curves indicate that the CC exceeded the 95% confidence level. A and C represent anti-cyclone and cyclone, respectively.**



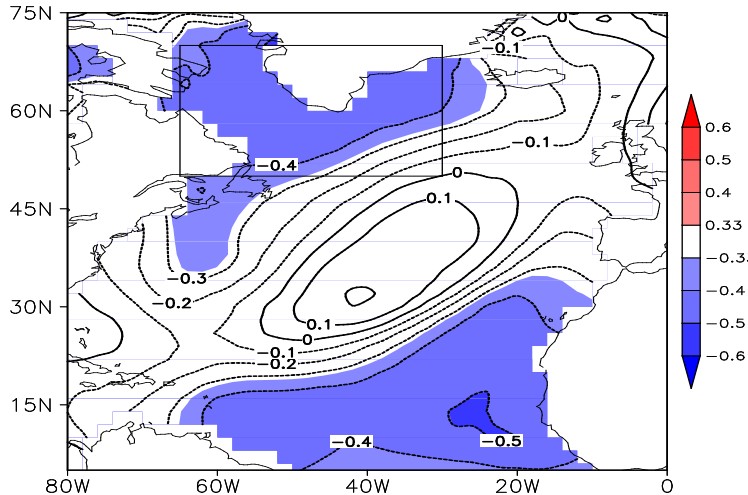

**Figure 6.** The CC between WHD$_{NCP}$ DY and Atlantic SST DY in autumn from 1980 to 2013. The shades indicate that the CC exceeded the 95% confidence level, and the rectangle represents the selected region (50–70°N,30–65°W) of predictor $x_3$.

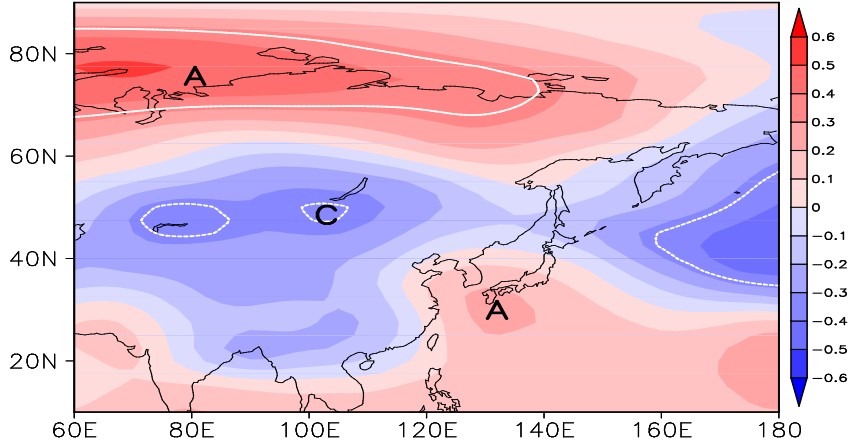

**Figure 7.** The CC between predictor $x_3$ (×−1) and Z500 DY in winter from 1980 to 2013. The white curves indicate that the CC exceeded the 95% confidence level. A and C represent anti-cyclone and cyclone, respectively.





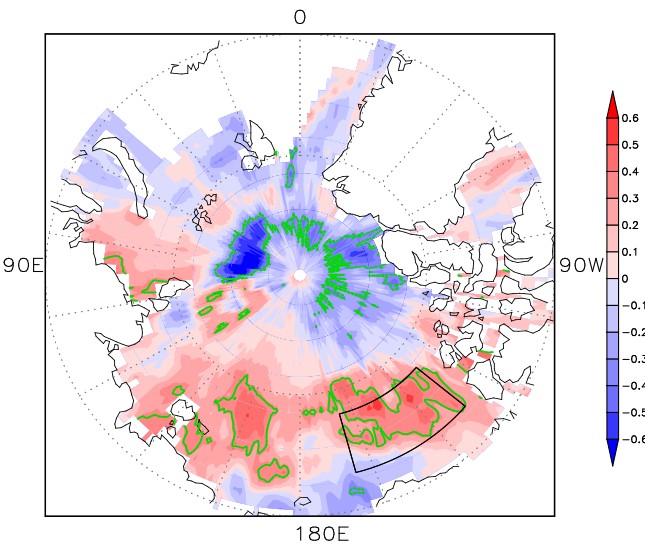

**Figure 8. The CC between WHD$_{NCP}$ DY and ASI DY in autumn from 1980 to 2013. The shades indicate that the CC exceeded the 95% confidence level, and the rectangle represents the selected region (73–78°N,130–165°W) of predictor $x_4$.**

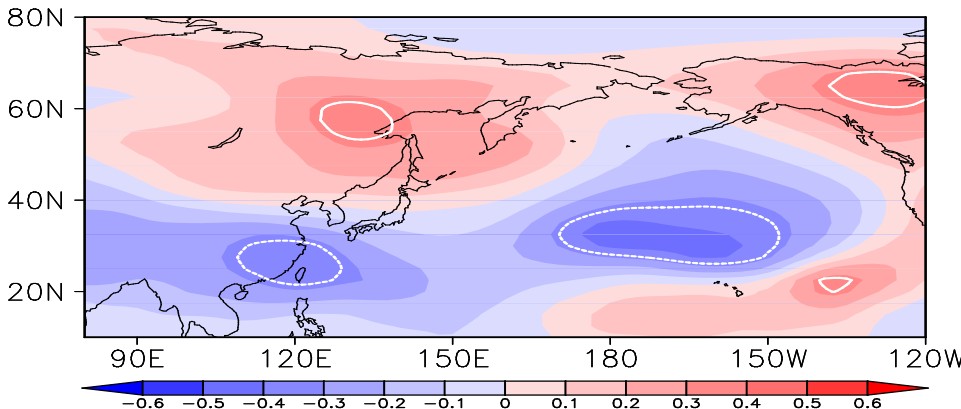

**Figure 9. The CC between predictor $x_4$ and Z500 DY in winter from 1980 to 2013. The white curves indicate that the CC exceeded the 95% confidence level.**





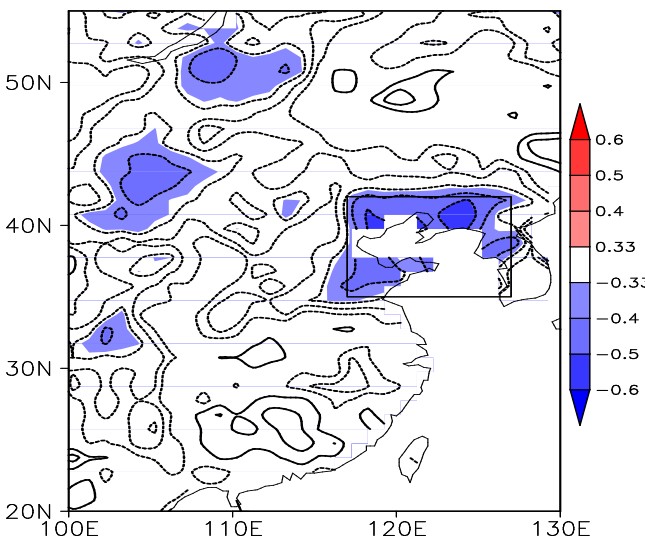

**Figure 10. The CC between WHD$_{NCP}$ DY and SoilM DY in autumn from 1980 to 2013. The shades indicate that the CC exceeded the 95% confidence level, and the rectangle represents the selected region (35–42$^o$N,117–127$^o$E) of predictor $x_5$.**

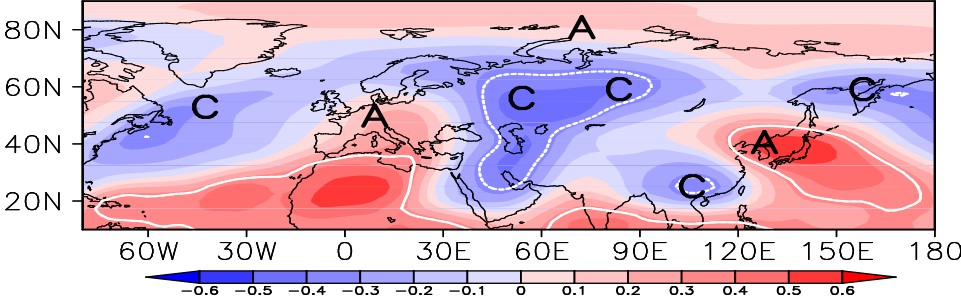

**Figure 11. The CC between predictor $x_5$ (×−1) and Z500 DY in winter from 1980 to 2013. The white curves indicate that the CC exceeded the 95% confidence level. A and C represent anti-cyclone and cyclone, respectively.**





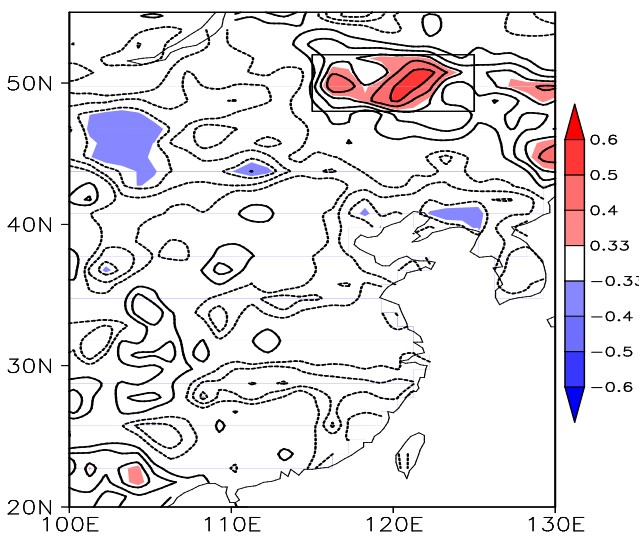

**Figure 12. The CC between WHD$_{NCP}$ DY and SoilM DY in summer from 1980 to 2013. The shades indicate that the CC exceeded the 95% confidence level, and the rectangle represents the selected region (48–52$^o$N, 115–125$^o$E) of predictor $x_6$.**

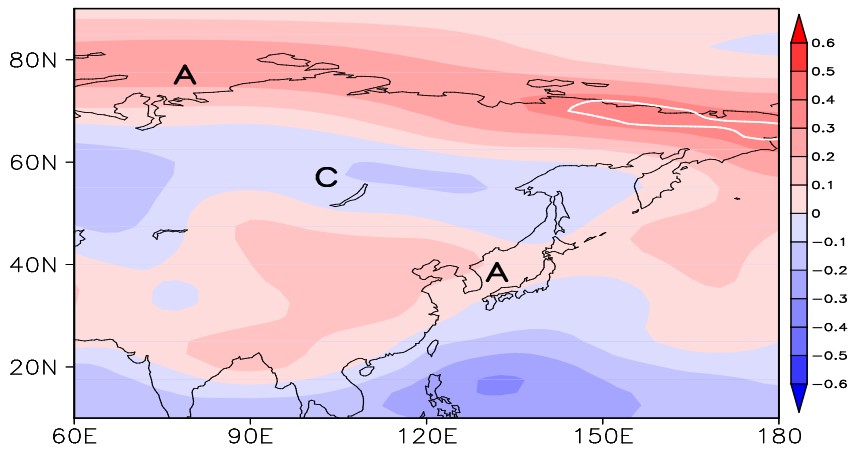


**Figure 13. The CC between predictor $x_6$ and Z500 DY in winter from 1980 to 2013. The white curves indicate that the CC exceeded the 95% confidence level. A and C represent anti-cyclone and cyclone, respectively.**



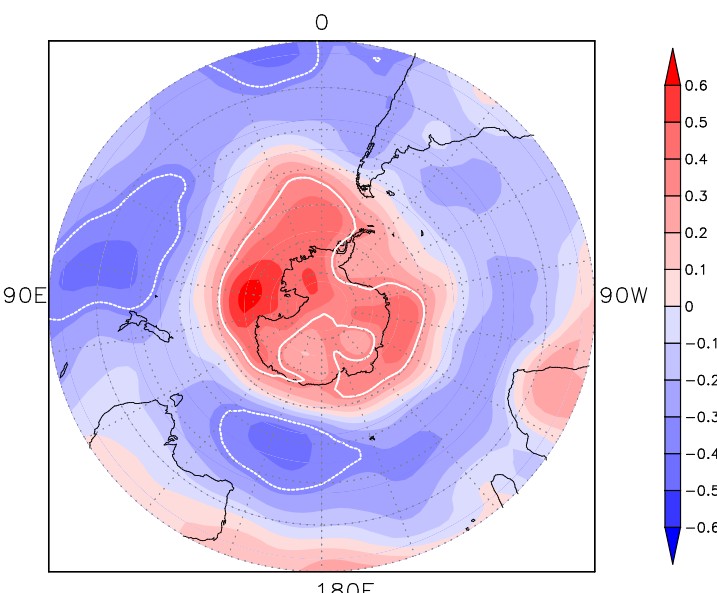

**Figure 14. The CC between WHD$_{NCP}$ DY and Sep–Oct Z850 DY from 1980 to 2013. The white curves indicate that the CC exceeded the 95% confidence level.**


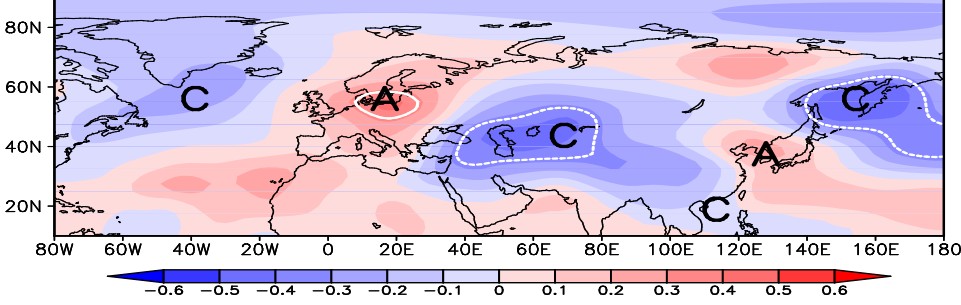

**Figure 15. The CC between predictor $x_7$ (×−1) and Z500 DY in winter from 1980 to 2013. The white curves indicate that the CC exceeded the 95% confidence level.**



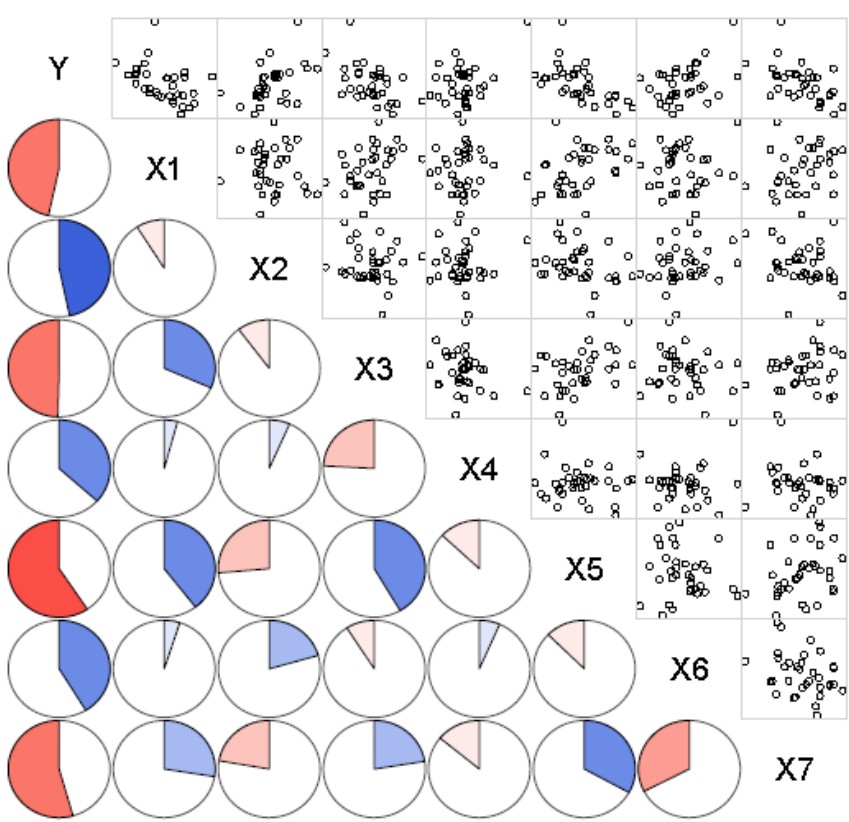

**Figure 16. Correlogram of the dependent (Y) and independent ($x_1, x_2 ..., and \ x_7$) variables, whose names were written on the diagonal. The lower panel shows the pie charts of correlation coefficients, representing the values by area and saturation, and showing positive/negative sign by blue/red, respectively. The upper panel shows the scatter plots.**



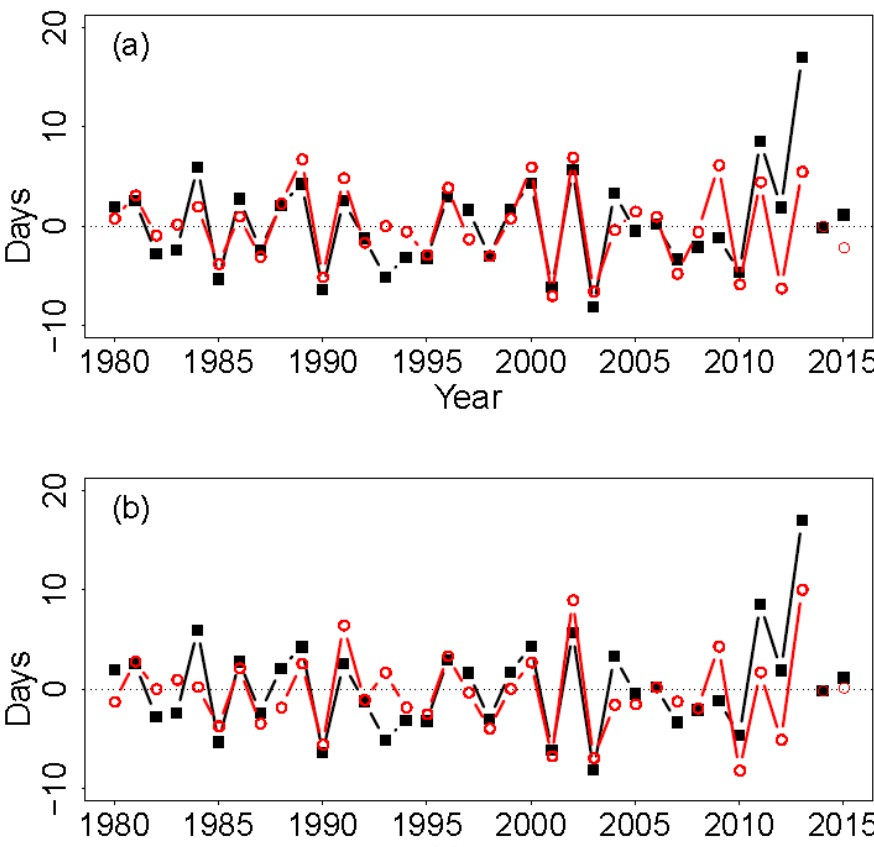

**Figure 17. The temporal variation of measured (black) WHD$_{NCP}$ DY, MLR (red, a) and GAM (red, b) cross-validation fitted WHD$_{NCP}$ DY from 1980 to 2013. The results for 2014 and 2015 represent the measured (black square) and predicted (red hollow circle) WHD$_{NCP}$ DY.**





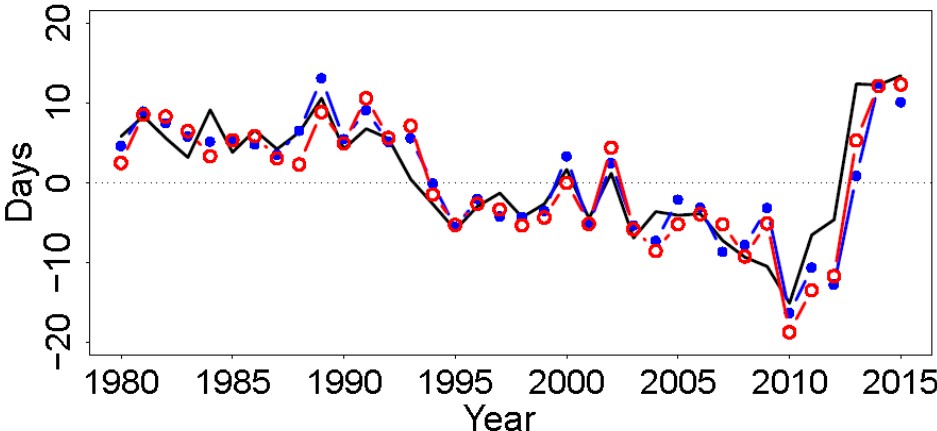

**Figure 18. The temporal variation of measured (black) WHD$_{NCP}$ anomaly from 1980 to 2015, MLR (blue) and GAM (red)**

**simulative WHD$_{NCP}$ anomaly, which was composed of cross fitted series from 1980 to 2013 and predicted values in 2014 and 2015.**