# Peer review of "Seasonal Prediction of Winter Haze Days in the North-Central North China Plain"

_Atmospheric Chemistry and Physics, 2016_

## Referee Comment (RC1) · Anonymous Referee #1 · 12 Sep 2016

General Comments:

The paper entitled "Seasonal prediction of winter haze days in the north-central North China Plain" selected seven predictors based on the analysis, and then used them to establish two statistical schemes for the prediction of the winter haze days over the north-central North China Plain. The two prediction models were demonstrated to have good ability to capture the interannual and interdecadal trends and the extremums of the haze days over the north-central North China. Thus, this study provides a good basis for the prediction of the haze days. I recommend the manuscript a minor revision before it be published in the journal.

Specific Comments:

(1) As is known, the human activity, especially the energy consumption, is the first

driver for the increasing of the haze days in North China in recent years, while the climatic conditions may be the second driver. So, how to take the human influence into account in the current prediction models? Some discussions about this issue are suggested to be included in the study.

(2) The predictors are selected mainly based on the correlation analysis. The correlations may indicate some relations (phenomena) but do not really imply causality (reason). To confirm the reliability of the selected predictors for the prediction models, the physical mechanisms underlying their relationships are suggested to be presented.

(3) There are far too many acronyms. This made me very confused when reading this paper. So, how can the authors expect the readers to remember all these acronyms in reading through the paper?

(4) The writing are needed to be further improved, and some sentences are needed to be rephrased to make them more clear. At several places, I cannot understand what is being conveyed. For example, "some new climatic finding have been helpful seasonal. . .", "soil moisture is an important . . . but only after SST", and so on. Please re-edit before submitting.

(5)Some references are cited in the text but not listed in the Reference section. Please check throughout the manuscript.

---

## Referee Comment (RC2) · Anonymous Referee #2 · 21 Sep 2016

General comments:

By using the year-to-year increment as the predictands, the authors established a statistical model to predict the winter haze pollution over the North-Central North China Plain (NCP), in which seven predictors are selected and two schemes are employed. Cross validation shows that such model can successfully capture the interannual and interdecadal variabilities of winter haze days over the NCP and the extremums as well. The model based on the new approach of year-to-year increment is very skillful for the seasonal prediction of winter haze days in the NCP, and has greatly potential applications to the environmental pollutions. However, more discussions are needed for the predictor selections and their possible physical processes in the successful seasonal prediction of winter haze days, so that the readers can better understand and apply this seasonal prediction model. Minor revision is required before it is accepted

for publication.

Several major issues need to be addressed:

1. Line 45: "climate change" usually indicates a long-term climate variation, but DY is more like interannual variability, so it's not appropriate to use the phrase "climate change" here.

2. Line 73-74 and Line 77: Why are the MLR and GLM called model-driven and data-driven methods, respectively? A brief description is highly encouraged.

3. Line 87, Line 97, Line 103 and many others: it should be noted that Pacific Japan (PJ) pattern is a summer teleconnection identified by Nitta (1987), not a winter one.

4. Line 91-92: Why is the water vapor transportation enhanced?

5. Line 95: Region of the Japan Sea to the Stanovoy Range is chosen for the pre-autumn TS DY, however, from Figure 2, we can see the region from the Japan Sea to the south of Lake Baikal has larger correlation coefficient. Maybe it is better to use this region for predictor x1.

6. Line 97: As defined by Wallace and Gutzler (1981), EU pattern has three major nodes with their locations at (55N, 20E), (55N, 75E) and (40N, 145E). However, Figure 3 doesn't cover the whole area of EU pattern, so we cannot obviously see the negative EU features from Figure 3. Besides, Figure 3 has rather different features from Figure 1 except for the anticyclone anomalies over the South Japan.

7. Line 103, Line 107: As for the PJ and negative EU pattern, they are not very clearly and significantly seen.

8. Line 110-113, Line 128-129: The correlation coefficients near the NCP are almost insignificant, how can the predictors x3 and x6 be so important?

9. Line 143: Figure 16 is a little complicated, it's better to explain it briefly.

10. Line 154: How to define the predicted bias?

11. Line 180: The center of positive geopotential anomalies is actually located over the Japan Sea.

12. Line 182-184: The consistence of SPMMLR and SPMGAM indicates that the linear part dominates the WHDNCP predictions. At the same time, the failure of predicting the rapid rising trend after 2010 also implies that the DY method has some deficiencies in dealing with the large abrupt change. The authors should point them out.

Technical corrections:

13. Line 25-26: The sentence is a little awkward, and it needs modification.

14. Line 38: The citation of Huang et al. 2015 is not present in the reference list.

15. Line 48: The citation of Huang et al. 2014 is not present in the reference list.

16. Line 91: "Asia" should be "Asian".

17. Line 105: As for the "Prior studies", some citations should be given.

18. Line 118-119: it needs to make clear what this sentence is talking about, geopotential height?

19. Line 119: "EAJS" should be "EASJ".

20. Line 160: "processing" should be "process".

21. Line 165: "results that are" can be changed to "and the results are".

22. Line 171: "simulative" to "simulated".

23. Keep the tense consistent in the whole paper.

---

## Author Comment (AC1) · 19 Oct 2016

**Reply letter to the anonymous Referee #1**

General Comments:

The paper entitled "Seasonal prediction of winter haze days in the north-central North China Plain" selected seven predictors based on the analysis, and then used them to establish two statistical schemes for the prediction of the winter haze days over the north-central North China Plain. The two prediction models were demonstrated to have good ability to capture the interannual and interdecadal trends and the extremums of the haze days over the north-central North China. Thus, this study provides a good basis for the prediction of the haze days. I recommend the manuscript a minor revision before it be published in the journal.

Specific Comments:

**(1) As is known, the human activity, especially the energy consumption, is the first driver for the increasing of the haze days in North China in recent years, while the climatic conditions may be the second driver. So, how to take the human influence into account in the current prediction models? Some discussions about this issue are suggested to be included in the study.**

*Reply*:

There was no doubt that the human activities were the first driver and contributor for the increasing of haze days in China and should be taken into account, but it was quite difficult to gather the associated dataset. Our studies based on the assumption (or compromise) that the socio-economic component varied slowly between the current and previous year. Thus, the socio-economic terms could be neglected in the DY approach and were contained again by adding the previous measurement. Although the assumption was rough and simple, it indeed supports a way to the seasonal prediction of haze days. This compromise might be unsuitable in certain years when this pollutant emission proportion varied dramatically. Fortunately, the climate factors also contributed significantly and the developed models showed good

performance. In the last section, following the kindly advice of the referee, we discussed the ideal scheme that used the preceding autumn energy consumption as a predictor.

*Revision in the last paragraph:*

......At the same time, if the $SPM_{MLR}$ performed well in some years, the $SPM_{GAM}$ also showed good ability in these years, and *vice versa*. One possible reason could be that some useful factors, most notably the human activities, were not included here. There is no doubt that the human activities, especial the energy consumption, was the first driver for the increasing of haze pollution. In this paper, we simply assumed that the difference in pollutant emissions between current and previous years was very small and that the socio-economic component of $WHD_{NCP}$ varied slowly. This assumption could support the seasonal prediction of haze days in most of the years, but still was a compromise. In certain years, especially the recent years, this pollutant emission proportion varied rapidly that needed to be taken into account. The preceding autumn energy consumption should be a good choice, but difficult to be measured, and its DY could be introduced into the developed models directly to improve the predictive skill.

**(2) The predictors are selected mainly based on the correlation analysis. The correlations may indicate some relations (phenomena) but do not really imply causality (reason). To confirm the reliability of the selected predictors for the prediction models, the physical mechanisms underlying their relationships are suggested to be presented.**

*Reply*:

Actually, the studies about the associated physical mechanism, i.e., how the external forcings influenced haze pollutions, were new and still insufficient. In this paper, we selected 7 predictors and could not present the physical mechanism that each external forcing stimulated such associated circulations. Following the suggestions, we cited the latest reference about the impact of Pacific and Atlantic SST and the ASI on haze

pollution, and also added some content and Figures about the way that the associated circulations impacted the WHD$_{NCP}$ DY. Finally, we pointed out that the underlying physical mechanism about the external forcing needed further and deeper studies and some useful hints could be found in this paper.

*Revision for each predictor:*

**For Predictor $x_1$, the following contents were revised:**

The features of negative EU and positive WP pattern could be identified clearly and the anomalous cyclone over South China and South China Sea was significant in the circulations associated with predictor $x_1$ (×−1) (Figure 3). Although the associated land-air interaction, especially in the DY field, was complicate and still unclear, according to the analysis of Figure 1, the horizontal and vertical diffusion of pollutant particles would be restricted efficiently.

**For Predictor $x_2$, the following contents were revised and Figure 5 was replaced:**

Chen et al. (Chen et al. 2015) found that the severe winter haze events in the North China were closely related with the weaker and northward EAJS. The positive SST DY around the Alaska Gulf could induce obviously anomalous cyclone over eastern China and the adjacent ocean, and the stimulated easterly weakened the core of EAJS. Furthermore, there was significantly anomalous southerly at the high latitude that restricted the cold activities from their source region and intensified the haze pollution over NCP (Figure 5).

[Figure]

Figure 5. The CC between predictor $x_2$ and wind vector DY at 200 hPa in winter from 1980 to 2013. The shade indicates that the CC between the zonal wind DY and $x_2$ exceeded the 95% confidence level.

**For Predictor $x_3$, the following contents were revised, Figure 7 was replaced and a latest reference was cited:**

Xiao et al (Xiao et al. 2015) proved the SST anomalies over the North Atlantic from summer to the following winter exhibit a significant relationship with winter haze days on both decadal and interannual timescale.

The most obvious DY atmospheric circulations related with predictor $x_3$ ($\times-1$) were the positive WP pattern, whose south center linked with a subtropical high (Figure 7). The continental high and marine low was both weakened by the anomalous geopotential height form the lower to middle layer that led to weaker EAWM and weaker cold air. The pressure gradient over the east coast of China also resulted in significant southerly anomalies, indicating smaller surface wind and more moisture and resulting in more WHD$_{NCP}$.

[Figure]

Figure 7. The CC between predictor $x_3$ ($\times-1$) and Z500 DY (shade)/850 hPa wind DY (arrow) in winter from 1980 to 2013. The dots indicate that the CC exceeded the 95% confidence level. A and C represent anti-cyclone and cyclone, respectively.

**For Predictor $x_4$, the following contents were revised:**

Thus, the EAJS was weakened by the induced easterly and shifted northward that illustrated less cold activities over NCP (Yang et al. 2002) and generated more haze days.

**For Predictor $x_5$, the following contents were revised:**

Following SST, the soil moisture is another important factor for seasonal prediction (Guo et al. 2007). The WHD$_{NCP}$ was closely correlated with the moisture conditions due to the hygroscopicity of the atmospheric particles (Yin et al. 2015a). ……Being specific to local circulations, the cyclone over South China and the anti-cyclone over NCP and West Pacific stimulated significant southeaster between them (Figure omitted) that transported more moisture but decelerated the surface wind in the NCP.

**For Predictor $x_6$, the following contents and Figure 13 were revised:**

The anomalous geopotential height was distributed zonally at high latitude indicating that the meridional circulations that transported cold air were weak. The positive high over NCP could confine the vertical motion and the vertical diffusion of atmospheric particles and intensify the haze pollution over the NCP.

[Figure]

Figure 13. The CC between predictor $x_6$ and Z500 DY in winter from 1980 to 2013. The white curves indicate that the CC exceeded the 95% confidence level. A and C represent anti-cyclone and cyclone, respectively.

**For Predictor $x_7$, the following contents and were revised:**

……The anomalous anti-cyclone over NCP and adjacent ocean not only led to stable atmosphere but also resulted in small wind and high humidity…….

**(3) There are far too many acronyms. This made me very confused when reading this paper. So, how can the authors expect the readers to remember all these acronyms in reading through the paper?**

*Reply*

Yes, too many acronyms make the article difficult to read.

*Revision in the last paragraph:*

We have deleted the acronyms that used less than (including) thrice, such as Z, SWP, PSS, TBO, CPC, GLM and EV.

**(4) The writing are needed to be further improved, and some sentences are needed to be rephrased to make them more clear. At several places, I cannot understand what is being conveyed. For example, "some new climatic finding have been helpful seasonal. . .", "soil moisture is an important . . . but only after SST", and so on. Please re-edit before submitting**

*Reply*

We have re-edited and improved the writing sentence by sentence, including the issues mentioned here and some others.

*Revision related to the issues mentioned here, and the other revisions were addressed in the manuscript.*

Some new climatic studies should be helpful for diagnosing seasonal predictors of winter haze days over the NCP (WHD$_{NCP}$)

Following SST, the soil moisture is another important factor for seasonal prediction (Guo et al. 2007).

(5)Some references are cited in the text but not listed in the Reference section. Please check throughout the manuscript.

*Reply*

We checked throughout the manuscript and added the missed and some latest references.

*Revisions:*

The accessorial references were listed below.

Czaja A, Frankignoul C. 1999. Influence of the North Atlantic SST onthe atmospheric circulation. Geophys. Res. Lett. 26: 2969 –2972

Huang Y Y, Wang H J, Fan K. 2014. Improving the Prediction of the Summer Asian-Pacific Oscillation Using the Interannual Increment Approach. J. Climate, 27: 8126—8134, doi: http://dx.doi.org/10.1175/JCLI-D-14-00209.1

Yang S, Lau K M, Kim K M. 2002. ariations of the East Asian Jet Stream and Asian–Pacific–American Winter Climate Anomalies. Journal of Climate, 15(3): 306—325

Yee T, Mitchell N. 1991. Generalized additive models in plant ecology, Journal of Vegetation Science, 2(5): 587—602

Xiao D, Li Y, Fan S J, Zhang R H, Sun J R, Wang Y. 2015. Plausible influence of Atlantic Ocean SST anomalies on winter haze in China. Theor. Appl. Climatol, 122: 249—257

---

## Author Comment (AC2) · 19 Oct 2016

**Reply letter to the anonymous Referee #2**

General comments:

By using the year-to-year increment as the predictands, the authors established a statistical model to predict the winter haze pollution over the North-Central North China Plain (NCP), in which seven predictors are selected and two schemes are employed. Cross validation shows that such model can successfully capture the interannual and interdecadal variabilities of winter haze days over the NCP and the extremums as well. The model based on the new approach of year-to-year increment is very skillful for the seasonal prediction of winter haze days in the NCP, and has greatly potential applications to the environmental pollutions. **However, more discussions are needed for the predictor selections and their possible physical processes in the successful seasonal prediction of winter haze days, so that the readers can better understand and apply this seasonal prediction model.** Minor revision is required before it is accepted.

*Reply*:

Actually, the studies about the associated physical mechanism, i.e., how the external forcings influenced haze pollutions, were new and still insufficient. In this paper, we selected 7 predictors and could not present the physical mechanism that each external forcing stimulated such associated circulations. Following the suggestions, we cited the latest reference about the impact of Pacific and Atlantic SST and the ASI on haze pollution, and also added some content and Figures about the way that the associated circulations impacted the $WHD_{NCP}$ DY. Finally, we pointed out that the underlying physical mechanism about the external forcing needed further and deeper studies and some useful hints could be found in this paper.

*General Revision and specific revision for x4, x5 and x7 (the specific revision for the other predictors would be presented under the specific issue):*

In the last discussion section, we discussed the way that the external forcings impacted the $WHD_{NCP}$ DY.

……Actually, the studies about the associated physical mechanism, i.e., how the external forcings influenced haze pollutions, were new and still insufficient. In this paper, the underlying physical process was presented mostly from the way that the associated circulations impacted the $WHD_{NCP}$ DY. Thus, the physical mechanism that the external forcings stimulated such associated circulations needed further studies…..

**For Predictor $x_4$, the following contents were revised:**

Thus, the EAJS was weakened by the induced easterly and shifted northward that illustrated less cold activities over NCP (Yang et al. 2002) and generated more haze days.

**For Predictor $x_5$, the following contents were revised:**

Following SST, the soil moisture is another important factor for seasonal prediction (Guo et al. 2007). The $WHD_{NCP}$ was closely correlated with the moisture conditions due to the hygroscopicity of the atmospheric particles (Yin et al. 2015a). ……Being specific to local circulations, the cyclone over South China and the anti-cyclone over NCP and West Pacific stimulated significant southeaster between them (Figure omitted) that transported more moisture but decelerated the surface wind in the NCP.

**For Predictor $x_7$, the following contents and were revised:**

……The anomalous anti-cyclone over NCP and adjacent ocean not only led to stable atmosphere but also resulted in small wind and high humidity…….

For publication, several major issues need to be addressed:

**1. Line 45: "climate change" usually indicates a long-term climate variation, but DY is more like interannual variability, so it's not appropriate to use the phrase "climate change" here.**

*Reply*:

Yes, this detail indeed indicates different connotation.

*Revision:*

We changed the phrase to "climate variability"

**2. Line 73-74 and Line 77: Why are the MLR and GLM called model-driven and datadriven methods, respectively? A brief description is highly encouraged.**

*Reply*:

A brief description and related reference was supplemented.

*Revision:*

The GAM is data-driven rather than model-driven. The resulting fitted values do not come from an apriori model that was adopted by MLR and generalized linear model. The rationale behind fitting a nonparametric model is that the structure of data should be examined first to choose an appropriate smooth function for each predictor; that is, the GAM allow the data to determine the shape of the smooth function (Yee et al. 1991).

**3. Line 87, Line 97, Line 103 and many others: it should be noted that Pacific Japan (PJ) pattern is a summer teleconnection identified by Nitta (1987), not a winter one.**

*Reply*:

Thank you for the reminder. The WP pattern should be accurate following the description from the CPC's website.

The WP pattern is a primary mode of low-frequency variability over the North Pacific **in all months**. During winter and spring, the pattern consists of a north-south dipole of anomalies, with one center located over the Kamchatka Peninsula and another broad center of opposite sign covering portions of southeastern Asia and the western subtropical North Pacific.

*Revision:*

The PJ pattern has been changed into WP pattern, and the anomalous cyclone over South China Sea was analyzed as a individual system. The particular revision could be seen in the revisions related the predictors.

**4. Line 91-92: Why is the water vapor transportation enhanced?**

*Reply*:

The reason was added in the paper.

*Revision:*

……Meanwhile, the water vapor transportation was also enhanced by anomalous southeaster in the lower troposphere (Figure omitted), creating favorable conditions for more WHD$_{NCP}$ than in the previous year.

[Figure]

Figure S. The LCC between Z500 DY (shade)/surface wind DY (arrow) in winter and WDY from 1980 to 2013. The dots indicate LCC exceed the 90% confidence level.

**5. Line 95: Region of the Japan Sea to the Stanovoy Range is chosen for the preautumn TS DY, however, from Figure 2, we can see the region from the Japan Sea to the south of Lake Baikal has larger correlation coefficient. Maybe it is better to use this region for predictor x1.**

*Reply*:

Before submission, we tried the SAT DY from the Japan Sea to the south of Lake Baikal, but the performance was not as good as the selected one. We speculated that the reason might be that there were some internal interactions among the predictors.

The balance among predictors was also a key point needed to be considered.

*Revision:*

Leave the predictor x1 as it was.

**6. Line 97: As defined by Wallace and Gutzler (1981), EU pattern has three major nodes with their locations at (55N, 20E), (55N, 75E) and (40N, 145E). However, Figure 3 doesn't cover the whole area of EU pattern, so we cannot obviously see the negative EU features from Figure 3. Besides, Figure 3 has rather different features from Figure 1 except for the anticyclone anomalies over the South Japan.**

*Reply*:

In this article, we followed the definition of EU from Wang et al (2016) and plotted a small Figure. Actually, if we expand the plotted area, the typical EU pattern as mentioned could be identified clearly. Thus, we re-plotted Figure 3 and revised the words to include more analysis about the physical process.

Wang H J, He S P. 2015b. The North China/Northeastern Asia Severe Summer Drought in 2014. Journal of Climate. 28(17), 6667−668

*Revision:*

**For Predictor $x_1$, the following contents and Figure 3 were revised:**

The features of negative EU and positive WP pattern could be identified clearly and the anomalous cyclone over South China and South China Sea was significant in the circulations associated with predictor $x_1$ (×−1) (Figure 3). Although the associated land-air interaction, especially in the DY field, was complicate and still unclear, according to the analysis of Figure 1, the horizontal and vertical diffusion of pollutant particles would be restricted efficiently.

[Figure]

Figure 3. The CC between predictor $x_1$ ($\times$−1) and Z500 DY in winter from 1980 to 2013. The white curves indicate that the CC exceeded the 95% confidence level. A and C represent anti-cyclone and cyclone, respectively.

**7. Line 103, Line 107: As for the PJ and negative EU pattern, they are not very clearly and significantly seen.**

*Reply*:

Line 103 and 107 indicate the predictor x2 (Figure 5) and x3 (Figure 7), respectively. In the submitted manuscript, the EU and PJ patterns were not clear enough at the middle troposphere. Thus, we replaced the Figures and analyzed the associated wind at 200 hPa with x2 (EAJS) and the associated WP pattern (and southeaster over East China) with x3. After revision, the associated circulations were significant and the physical process was clearer.

*Revision:*

**For Predictor $x_2$, the following contents were revised and Figure 5 was replaced:**

Chen et al. (Chen et al. 2015) found that the severe winter haze events in the North China were closely related with the weaker and northward EAJS. The positive SST DY around the Alaska Gulf could induce obviously anomalous cyclone over eastern China and the adjacent ocean, and the stimulated easterly weakened the core of EAJS. Furthermore, there was significantly anomalous southerly at the high latitude that

restricted the cold activities from their source region and intensified the haze pollution over NCP (Figure 5).

[Figure]

Figure 5. The CC between predictor $x_2$ and wind vector DY at 200 hPa in winter from 1980 to 2013. The shade indicates that the CC between the zonal wind DY and $x_2$ exceeded the 95% confidence level.

**For Predictor $x_3$, the following contents were revised, Figure 7 was replaced and a latest reference was cited:**

Xiao et al (Xiao et al. 2015) proved the SST anomalies over the North Atlantic from summer to the following winter exhibit a significant relationship with winter haze days on both decadal and interannual timescale.

The most obvious DY atmospheric circulations related with predictor $x_3$ ($\times-1$) were the positive WP pattern, whose south center linked with a subtropical high (Figure 7). The continental high and marine low was both weakened by the anomalous geopotential height form the lower to middle layer that led to weaker EAWM and weaker cold air. The pressure gradient over the east coast of China also resulted in significant southerly anomalies, indicating smaller surface wind and more moisture and resulting in more WHD$_{\text{NCP}}$.

[Figure]

Figure 7. The CC between predictor $x_3$ ($\times-1$) and Z500 DY (shade)/850 hPa wind DY (arrow) in winter from 1980 to 2013. The dots indicate that the CC with meridional wind exceeded the 95% confidence level. A and C represent anti-cyclone and cyclone, respectively.

**8. Line 110-113, Line 128-129: The correlation coefficients near the NCP are almost insignificant, how can the predictors x3 and x6 be so important?**

*Reply*:

Line 110—113 and 128—129 related with the predictor x3 (Figure 7) and x6 (Figure 13), respectively. The issue about x3 was addressed in Q7. The southerly anomalies at the lower troposphere were significant (dots in Figure 7) indicating smaller surface wind and more moisture. For predictor x6, we expand the plotted area; the typical EU pattern could be identified, but the correlation coefficients near the NCP were still insignificant. In consideration of distributed pattern, we still kept this predictor. In addition to the teleconnection, the local meteorological circulation and conditions were analyzed, including the weaker meridional circulations and vertical motion.

*Revision:*

**For Predictor $x_6$, the following contents and Figure 13 were revised:**

The anomalous geopotential height was distributed zonally at high latitude indicating that the meridional circulations that transported cold air were weak. The positive high over NCP could confine the vertical motion and the vertical diffusion of atmospheric

particles and intensify the haze pollution over the NCP.

[Figure]

Figure 13. The CC between predictor $x_6$ and Z500 DY in winter from 1980 to 2013. The white curves indicate that the CC exceeded the 95% confidence level. A and C represent anti-cyclone and cyclone, respectively.

**9. Line 143: Figure 16 is a little complicated, it's better to explain it briefly.**

*Reply*:

Yes, Figure 16 is a little complicated and the information exceeded the necessary demand. In another word, the texts were enough. What we wanted to presented was that only 5 pairs of the predictors showed significant linear correlation and the multicollinearity would not be a problem when modeling with the MLR approach.

*Revision:*

Figure 16 was deleted.

**10. Line 154: How to define the predicted bias?**

*Reply*:

The definition was supplemented.

*Revision:*

……As an independent prediction test, the predicted bias, i.e., the predicted value minus the measurement, in 2014 was 0.09……

**11. Line 180: The center of positive geopotential anomalies is actually located over the Japan Sea.**

*Reply*:

The description was revised.

*Revision:*

……positive anomalies over the NCP and Japan Sea…….

**12. Line 182-184: The consistence of SPMMLR and SPMGAM indicates that the linear part dominates the WHDNCP predictions. At the same time, the failure of predicting the rapid rising trend after 2010 also implies that the DY method has some deficiencies in dealing with the large abrupt change. The authors should point them out.**

*Reply*:

The reminder from the referee was important for the discussion. Thus, we enriched the discussion about the comparison between MLR and GAM, and the DY approach itself.

*Revision:*

……The consistence of these two models might indicate that, after including plentiful predictors, the linear relationship dominated the $WHD_{NCP}$ DY prediction…….

……The large abrupt change was a common challenge to the statistical models, including the DY approach, so the numerical model should be introduced into the prediction of haze pollution. At the same time, if the $SPM_{MLR}$ performed well in some years, the $SPM_{GAM}$ also showed good ability in these years, and *vice versa*. One possible reason could be that some useful factors, most notably the human activities, were not included here. There is no doubt that the human activities, especial the energy consumption, was the first driver for the increasing of haze pollution. In this paper, we simply assumed that the difference in pollutant emissions between current

and previous years was very small and that the socio-economic component of WHD$_{NCP}$ varied slowly. This assumption could support the seasonal prediction of haze days in most of the years, but still was a compromise. In certain years, especially the recent years, this pollutant emission proportion varied rapidly that needed to be taken into account. The preceding autumn energy consumption should be a good choice, but difficult to be measured, and its DY could be introduced into the developed models directly to improve the predictive skill…….

**Technical corrections:**

**13. Line 25-26: The sentence is a little awkward, and it needs modification.**

*Reply*:

The sentence has been corrected.

*Revision:*

Some new climatic studies should be helpful for diagnosing seasonal predictors of winter haze days over the NCP (WHD$_{NCP}$)

**14. Line 38: The citation of Huang et al. 2015 is not present in the reference list.**

*Reply*:

This citation has been merged into Huang et al. 2014

*Revision:*

……the signals (i.e., variance) of the predictors and predictand were both amplified (Huang et al. 2014) and……

**15. Line 48: The citation of Huang et al. 2014 is not present in the reference list.**

*Reply*:

This citation of Huang et al. 2014 has been listed in the reference.

*Revision:*

Huang Y Y, Wang H J, Fan K. 2014. Improving the Prediction of the Summer Asian-Pacific Oscillation Using the Interannual Increment Approach. J. Climate, 27: 8126—8134, doi: http://dx.doi.org/10.1175/JCLI-D-14-00209.1

**16. Line 91: "Asia" should be "Asian".**

*Reply*:

The error has been corrected.

*Revision:*

……together with the cyclone; they could induce an easterly to weaken the East Asian Jet Stream (EAJS), producing weaker cold air.

**17. Line 105: As for the "Prior studies", some citations should be given.**

*Reply*:

The citation below was given.

Czaja A, Frankignoul C. 1999. Influence of the North Atlantic SST onthe atmospheric circulation. Geophys. Res. Lett. 26: 2969 –2972

*Revision:*

Prior studies have documented that the triple SST pattern was a dominant mode of the northern Atlantic SST in autumn (Czaja et al. 1999).

**18. Line 118-119: it needs to make clear what this sentence is talking about, geopotential height?**

*Reply*:

Yes, it was talking about geopotential height and has been rewritten.

*Revision:*

……The continental high and marine low was both weakened by the anomalous geopotential height form the lower to middle layer that led to weaker EAWM and

weaker cold air…….

**19. Line 119: "EAJS" should be "EASJ".**

*Reply*:

We made a mistake when defining the acronym. It should be East Asian Jet Stream (EAJS), and we corrected them throughout the paper.

*Revision:*

……together with the cyclone; they could induce an easterly to weaken the East Asian Jet Stream (EAJS), producing weaker cold air…….

**20. Line 160: "processing" should be "process".**

*Reply*:

The error has been corrected.

*Revision:*

……and show the trend well, indicating an advantage to process the non-linear relationship……

**21. Line 165: "results that are" can be changed to "and the results are".**

*Reply*:

The error has been corrected.

*Revision:*

The predicted bias in 2014 and 2015 was −0.07 and −1.01, and the results are slightly better than those from $SPM_{MLR}$.

**22. Line 171: "simulative" to "simulated".**

*Reply*:

The error has been corrected.

*Revision:*

In Figure 17, the simulated WHD$_{NCP}$ anomaly was fitted by cross-validation from 1980 to 2013 and predicted in 2014 and 2015.

**23. Keep the tense consistent in the whole paper.**

*Reply*:

The tense has been checked throughout and, then was kept consistent.